# Super-resolution spectroscopic microscopy via photon localization

Biqin Dong[1,2,*], Luay Almassalha[1,*], Ben E. Urban[1,*], The-Quyen Nguyen[1,*], Satya Khuon[3], Teng-Leong Chew[3], Vadim Backman[1], Cheng Sun[2] & Hao F. Zhang[1]

Traditional photon localization microscopy analyses only the spatial distributions of photons emitted by individual molecules to reconstruct super-resolution optical images. Unfortunately, however, the highly valuable spectroscopic information from these photons have been overlooked. Here we report a spectroscopic photon localization microscopy that is capable of capturing the inherent spectroscopic signatures of photons from individual stochastic radiation events. Spectroscopic photon localization microscopy achieved higher spatial resolution than traditional photon localization microscopy through spectral discrimination to identify the photons emitted from individual molecules. As a result, we resolved two fluorescent molecules, which were 15 nm apart, with the corresponding spatial resolution of 10 nm—a four-fold improvement over photon localization microscopy. Using spectroscopic photon localization microscopy, we further demonstrated simultaneous multi-colour super-resolution imaging of microtubules and mitochondria in COS-7 cells and showed that background autofluorescence can be identified through its distinct emission spectra.

[1] Biomedical Engineering Department, Northwestern University, Evanston, Illinois 60208, USA. [2] Mechanical Engineering Department, Northwestern University, Evanston, Illinois 60208, USA. [3] Advanced Imaging Center, Howard Hughes Medical Institute, Ashburn, Virginia 20147, USA. * These authors contributed equally to this work. Correspondence and requests for materials should be addressed to H.F.Z. (email: hfzhang@northwestern.edu).

In Raman scattering and fluorescence excitation and emission, incident photons interact with the intrinsic electronic or vibrational states of the sample[1,2] and subsequently emit frequency-shifted photons due to the underlying energy exchange[3–6]. Analysing the spectroscopic signatures obtained from inelastic light scattering measurements is a widely used method for revealing the electronic and structural properties for natural and engineered materials in subjects ranging from biology to materials science[7–9]. In addition, a variety of spectroscopic imaging techniques have been developed to probe the heterogeneous environment within samples[10–12], yet their spatial resolutions have been limited to about half of the wavelength due to light diffraction. Although near-field scanning optical microscopy offers nanometre-scale spatial resolution by using a sharp stylus for scanning at the close vicinity of the sample surface[13,14], it is unable to image sub-surface features because rapidly decaying evanescent fields are accessible only within the optical near-field. Therefore, further development of far-field spectroscopic nanoscopy remains highly desirable.

Recent advancements in super-resolution fluorescence microscopy have extended the ultimate resolving power of far-field optical microscopy significantly beyond the diffraction limit. A wide range of imaging modalities, including structured illumination microscopy, stimulated emission depletion microscopy and photon localization microscopy (PLM), have been successfully developed[15–19]. In particular, PLM, which includes photoactivated localization microscopy (PALM) and stochastic optical reconstruction microscopy (STORM), relies on the stochastic radiation of individual fluorescent molecules to determine the probabilistic locations from their localized point spread functions while providing deep sub-diffraction-limited spatial resolution. Notably, PLM does not alter the emission spectrum of the stochastic radiation, making it promising for the development of a spectroscopic nanoscope. Previously, analysing the spectroscopic features of the stochastic radiation of single molecules has been demonstrated by recording multiple images from discrete wavelength bands[20]. However, due to the limited imaging sensor area of a single CCD camera, only several wavelength bands can be recorded simultaneously. The resulting poor spectral resolution makes this method unable to resolve fine spectral details and distinguish spectra with overlapping emission bands[21]. Further improvement of spectral resolution is limited to the overall size of the imaging sensor array, and it can be rather complicated and expensive if multiple cameras are employed.

Here we report spectroscopic photon localization microscopy (SPLM), a newly developed far-field spectroscopic imaging technique[22], which is capable of simultaneously capturing multiple molecular contrasts from individual molecules at nanoscopic scale. While similar optical configuration was recently reported in enabling multicolour super-resolution imaging in three-dimensional[23]; however, SPLM permits fluorescence spectral analysis of individual molecules with experimentally demonstrated sub-nanometre spectral resolution and sub-10 nanometre spatial resolution. Using a slit-less monochromator, both the zero-order and the first-order diffractions from a grating were recorded simultaneously to reveal the spatial distribution and the associated emission spectra of individual stochastic radiation events, respectively. Whereas conventional PLM analyses only the centroid of each stochastic radiation event, SPLM further captures and correlates the associated emission spectrum with the location of centroids. By employing spectral unmixing[11] and regression[24], individual fluorescent molecules located within the close proximity can be still distinguished according to their emission spectra. Taking advantage of this unique capability, SPLM significantly extends

the fundamental spatial resolution limit of photon localization microscopy by at least a factor of four through combining photons from same fluorophore. Our approach not only enhances existing super-resolution imaging by capturing the molecule-specific spectroscopic signatures, it will potentially provide a universal platform for unravelling heterogeneous nanoscale environments in complex systems at the single-molecule level.

## Results

**Principle of spectroscopic photon localization microscopy.** The working principle of SPLM is schematically illustrated in Fig. 1; the technical details are provided in the Methods section and in Supplementary Fig. 1. As shown in Fig. 1a, a continuous-wave laser illumination was used to excite the fluorescent molecules into long-lived dark states and subsequently recover them by stochastic photo-switching[17–19]. The resulting fluorescence image was coupled into a Czerny–Turner type monochromator (SP2150, Princeton Instruments) featuring a blazed dispersive grating (150 grooves per mm). The collected fluorescent emission was divided at an approximate 1:3 ratio between the zeroth and the first diffraction orders. To simultaneously acquire the zero-order and first-order images using a high-sensitivity EMCCD camera (proEM, Princeton Instruments), a mirror was placed in the monochromator to adjust the position of the zero-order image. This is a critical step for establishing the temporal and spatial correlations between zero-order and first-order images in dealing with the sparsely distributed emissions which are stochastic by nature.

We first investigated a dual-stained sample with mixed actin monomers (Cytoskeleton) labelled with Alexa Fluor 532 (Life Technologies) and Alexa Fluor 568 (Life Technologies). Our sample was excited using a 532-nm laser with a beam fluence of $2 \, \text{kW} \, \text{cm}^{-2}$. A movie containing a time sequence of 1000 image frames was recorded with a 20-ms exposure time for each frame. Each image frame contained simultaneously captured zero-order and first-order images. A captured frame shown in Fig. 1b is used as a representative example to illustrate the working principle of SPLM. The zero-order image does not impose additional dispersive characteristics of the grating, and thus it can be used to localize the positions of individual stochastic radiation events. The remaining photons are allocated to the first diffraction order to form a spatially dispersed image that reveals the fluorescence spectra of individual fluorescence dye molecules. For illustration, six stochastic radiation events and their corresponding emission spectra are numbered in Fig. 1b. Each stochastic radiation event originating from a single dye molecule remains spatially confined as a sub-diffraction-limited point source. Thus, it is possible to eliminate the need for a monochromator entrance slit without compromising the spectral resolution. This is particularly beneficial, as it allows for simultaneous acquisition of stochastic radiation events over a wide field-of-view and the associated fluorescent spectra.

Figure 1c shows a wide-field fluorescence image of the sample with diffraction-limited resolution. While PLM is capable of providing much improved spatial resolution by recording and analysing a movie containing sparsely distributed emissions (Fig. 1d), the lack of spectroscopic information makes PLM unable to distinguish different fluorescence dyes based on their spectroscopic signatures. In contrast, SPLM simultaneously records both spatial position and spectroscopic information. As shown in Fig. 1e, the zero-order images are analysed by using the standard localization algorithm[25] (QuickPALM, ImageJ plug-in) to determine the locations of individual blinking events, which is identical to the processing used in PALM and STORM. The mirror flipped the zero-order images horizontally, but this transformation was easily reversed during image

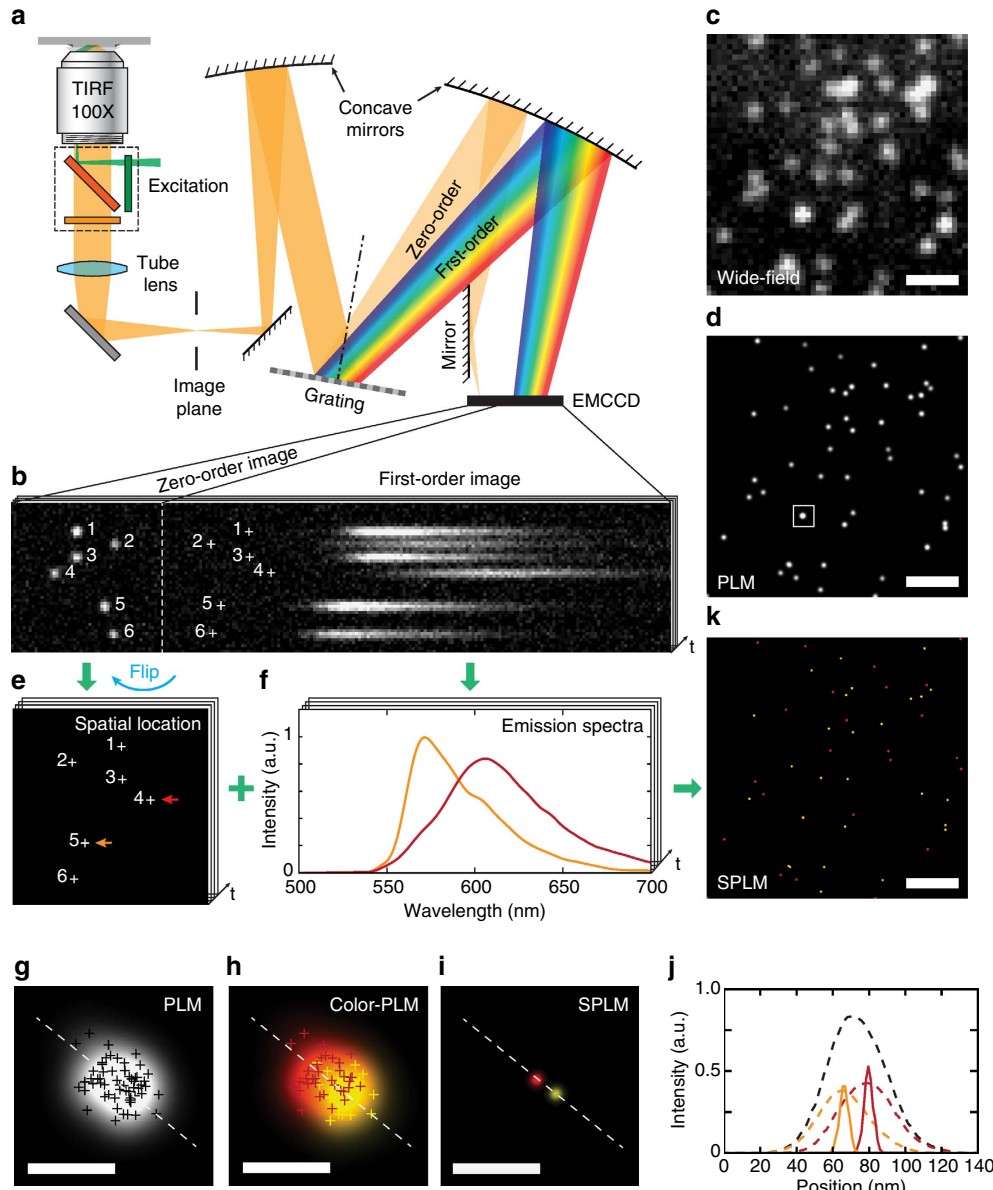

**Figure 1 | The working principle of SPLM.** (**a**) Schematic of the SPLM system. Upon laser excitation, the fluorescent image was collected by a high-numerical aperture objective lens and subsequently coupled into a Czerny–Turner monochromator by a match tube lens. (**b**) Both the zero-order and the first-order diffractions from the grating were recorded simultaneously using the same EMCCD camera. (**c**) Wide-field optical image of the sample consisting of actin monomers labelled by Alexa Fluor 532 and Alexa Fluor 568 with diffraction-limited resolution. (**d**) The conventional PLM method offers sub-diffraction-limited imaging resolution, but is unable to capture the spectroscopic signature of the individual emitter. Scale bars: 1 μm (**c**,**d**). (**e**) The localization algorithm was used to determine the spatial locations of each blinking, illustrated by numbered crosses. These locations can be further used as the inherent reference points for spectral calibration of the emission spectra in the first-order image, shown as denoted crosses in **b**. (**f**) Representative spectra from two individual blinking events (highlighted by the coloured arrows in **e**). (**g**) Magnified view of the square region in the PLM image (**d**). (**h**) Corresponding colour-coded image by separating the spectra of individual stochastic localizations according to the emission characterization of the two dyes. The spectral regression of nearby localizations indicates the cluster consisted of two single-dye molecules. (**i**) By averaging the nearby localizations, the SPLM image with spectral regression shows the localization precision of two molecules. Scale bars, 50 nm (**g**–**i**). (**j**) Line profiles were used to compare the localization precision of PLM (black dashed line), colour-PLM (coloured dashed lines) and SPLM (coloured solid lines). (**k**) The super-resolution spectroscopic image was obtained by combining the spatial and spectroscopic information from all localizations. Scale bar, 1 μm.

processing. The centroid position serves two roles: (1) to determine the location of each activated fluorescent molecule (shown as numbered crosses in Fig. 1e) and (2) to establish a reference point to the corresponding emission spectrum from the measured first-order image (shown as numbered crosses in Fig. 1b).

To obtain optimal spatial and spectral resolutions, background signals, such as auto-fluorescence, Raman, or Rayleigh scattering

from the sample, must be carefully removed. In this work, the background signals were removed by subtracting the average of the adjacent image frames without the presence of stochastic emission (Supplementary Fig. 2). The dispersion of the imaging system was calibrated prior to image acquisition (Supplementary Fig. 3). The recorded spectrum from first-order image was further normalized by the wavelength-dependent characteristic of optical components and EMCCD (Supplementary Fig. 4). Taking into

consideration the focal length of the monochromator, the dispersive characteristic of the grating, and the pixel size of the EMCCD camera, we achieved a spectral resolution of 0.63 nm per pixel from the recorded first-order image. Figure 1f shows the spectra from two individual blinking events, which match reasonably well with the emission spectra of fluorescent dyes being used: Alexa Fluor 532 and Alexa Fluor 568. In Fig. 1e, the arrows of matching colour highlight the corresponding spatial locations for the spectra. Given the sparse nature of the stochastic emissions, the measured spectra from neighbouring fluorescence molecules are unlikely to overlap in space. In the rare event in which overlap occurs, the spectra of neighbouring fluorescence molecules can be separated with a customized spectral unmixing algorithm (Supplementary Note 1 and Supplementary Figs 5–7).

**Improving spatial resolution with spectral regression.** The demonstrated capability of SPLM to distinguish minor differences in fluorescent spectra offers unique advantages compared with conventional PLM. Figure 1g shows a magnified view of the highlighted region in the conventional PLM image (Fig. 1d). Every localization event is convolved with a Gaussian kernel, where the full-width at half-maximum is determined by the localization precision[26]. It appears that the multiple stochastic emissions are clustered in close proximity. However, by examining the emission spectra from individual stochastic emissions using SPLM, we discovered that the emissions actually originated from two different types of fluorescent dye molecules (Supplementary Fig. 8). By colour-coding each localization event with its spectroscopic signature, we can determine that the centroids of two fluorescent molecules are spatially separated (Fig. 1h). Given the level of the dilution of the dye molecules, it is reasonable to expect that the observed clusters of stochastic emissions are originated from single-dye molecules. Using this knowledge, we applied a spectral regression algorithm to identify the emissions from the same dye molecules and

subsequently to accumulate all the photons for the localization analysis. With this technique, we were able to improve the localization precision to sub-10 nm. The two fluorescent molecules with 15 nm centre-to-centre spacing can be clearly distinguished (Fig. 1i,j).

Notably, since only one-fourth of the total photons emitted was allocated to the zero-order image, it resulted in a two-fold reduction in the spatial resolution according to the localization precision. Experimentally, we have observed ∼40 nm spatial resolution in this study, which suggests a theoretical resolution limit of ∼20 nm if all emitted photons were allocated to the zero-order image. By applying the spectral regression algorithm, we can combine the photons from the multiple stochastic emissions from the same fluorescent dye molecule to improve the imaging resolution. As an example shown in Fig. 1d, the recording of 23,924 localizations can be classified to 1,582 clusters, indicating an average repeated occurrence of 15.1 under the given imaging buffer and laser excitation conditions (see Methods). The resolution analysis based on localization precision shows the dramatically improved spatial resolution from 39.0 to 9.8 nm by applying spectral regression algorithm (Supplementary Fig. 9), which is more than a two-fold improvement over the standard PLM method. Nevertheless, the ultimate spatial resolution depends on the total number of photons emitted by individual dye molecules, which is eventually determined by the irreversible photo-bleaching threshold. Finally, super-resolution spectroscopic imaging can be accomplished, as illustrated in Fig. 1k.

Even for the same type of molecules, individual molecules can be differentiated by exploiting their heterogeneous fluorescence[5,9,13]. To verify this, we imaged actin monomers labelled solely with Alexa Fluor 568. Figure 2a shows a wide-field fluorescence image, and Fig. 2b shows the corresponding conventional PLM image. Figure 2c shows a magnified view of two clusters from the yellow box in Fig. 2b. The dye molecules can be repetitively activated, and their stochastic emissions can be

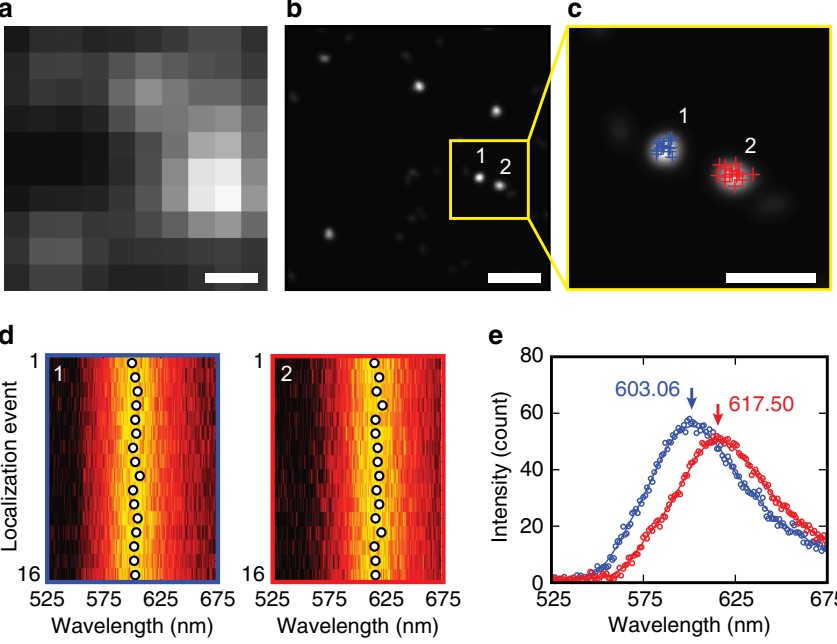

**Figure 2 | Differentiating individual molecules by exploiting their heterogeneous fluorescence.** (**a**) Wide-field optical image and (**b**) PLM image of actin monomers labelled by Alexa Fluor 568. PLM image was reconstructed from localization coordinates using localization precision as the full-width at half-maximum of a Gaussian kernel. (**c**) Two nearby clusters are highlighted and localization coordinates are marked by crosses. Scale bars, 200 nm (**a**,**b**); 100 nm (**c**). (**d**) Emission spectra denoted by coloured circles in **c**. (**e**) The corresponding averaged spectra of these two clusters showing distinct emission peaks.

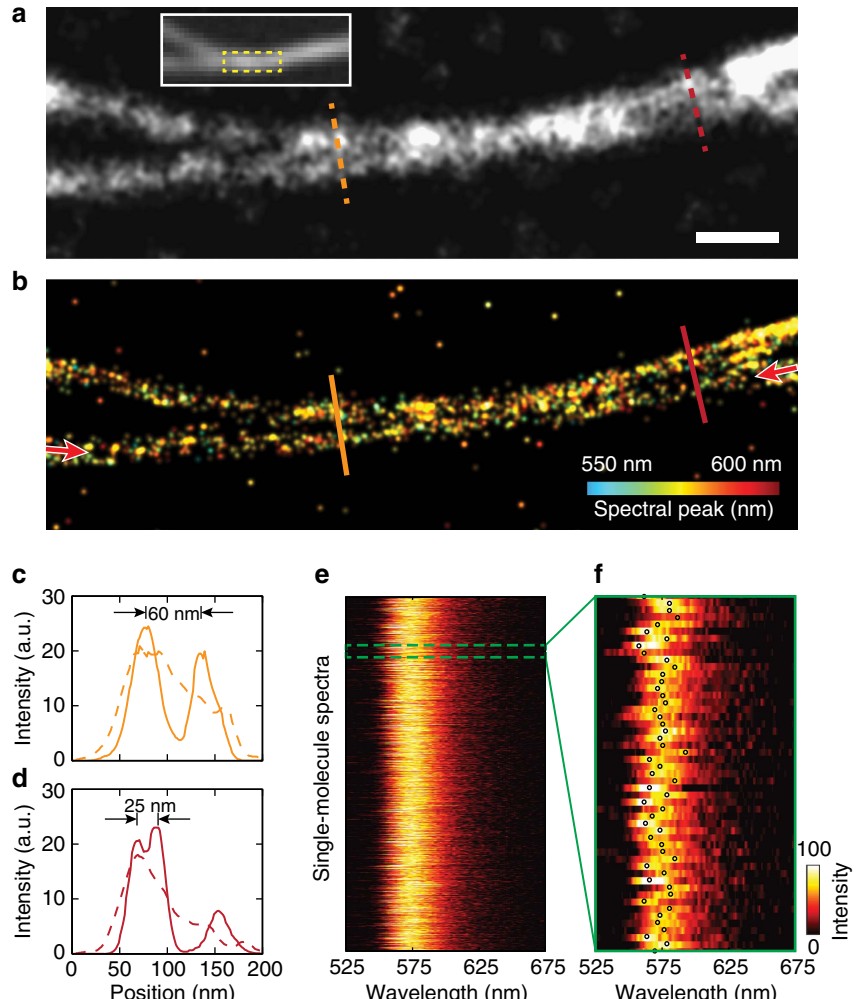

**Figure 3 | Imaging *ex vivo* microtubules using SPLM.** (**a**) Conventional PLM image of two closely spaced microtubules of the square region in the wide-field fluorescence image, as shown in the inset. Scale bar, 200 nm. (**b**) SPLM image with spectral regression. (**c,d**) are the line profiles from positions highlighted by the dashed and solid lines in **a,b**, respectively. (**e**) Emission spectra along a single microtubule, highlighted by the arrows in **b**. (**f**) Magnified view of the spectral variation. The circles indicate the peak positions of each spectrum.

recorded in multiple frames. The evolutions of their fluorescence spectra are shown in Fig. 2d (the spectra of other clusters in Fig. 2b are provided in Supplementary Fig. 10). Each individual cluster was found to exhibit repeatable emission spectra with a small variation in the peak position of 2.25 ± 0.45 nm. In contrast, different clusters exhibited clearly distinguishable spectra. As shown in Fig. 2e, the peak positions of the averaged fluorescence spectra for clusters #1 and #2 were 603.1 nm (s.d. = 1.8 nm) and 617.5 nm (s.d. = 2.1 nm), respectively, with a corresponding wavelength shift of 14.4 nm. The observed heterogeneous fluorescent spectra from molecules of the same type appear to be caused by molecular conformational variations and environmental heterogeneity[3,5,9,13,27,28]. These findings led us to believe that the multiple stochastic radiation events within a localized cluster were originating from the same dye molecules, which can be used to identify the origin of stochastic emission, judging by proximity in space and similarity in the emission spectra (see detailed methods and discussion in Supplementary Note 2 and Supplementary Figs 11 and 12).

**Super-resolution spectroscopic imaging of microtubules**. We further validated the improved SPLM imaging resolution using the Rhodamine-labelled microtubule samples (see sample preparation section in Methods). Figure 3a shows a conventional

PLM image of two closely spaced microtubules. After applying spectral regression, we rendered a SPLM image, as shown in Fig. 3b, with the colour representing the peak wavelength of each molecule. As shown by the line profiles in Fig. 3c, the two microtubules, which were difficult to distinguish from each other in the PLM image (dashed line), can be clearly resolved in the SPLM image (solid line). Features as small as 25 nm can be resolved from single microtubules (Fig. 3d). Figure 3e shows the spectra of all localization events in one of the microtubules, indicated by arrows in Fig. 3b. As illustrated in the magnified view (Fig. 3f), dye molecules of the same type have variations in spectral emissions due to the underlying fluorescent heterogeneity.

**Super-resolution spectroscopic cellular imaging**. Another crucial advantage of SPLM is that it may enable multi-label super-resolution imaging from a single round of acquisition. We demonstrated this advantage of SPLM by imaging dual-stained COS-7 cells. We used Alexa Fluor 568 and Mito-EOS 4b to stain the microtubules and mitochondria of a cell culture, respectively (see sample preparation section in Methods). Figure 4a,b shows a wide-field fluorescence image and a reconstructed SPLM image, respectively, of a dual-stained COS-7 cell. In the SPLM image, different colours represent the spectral peak wavelengths of

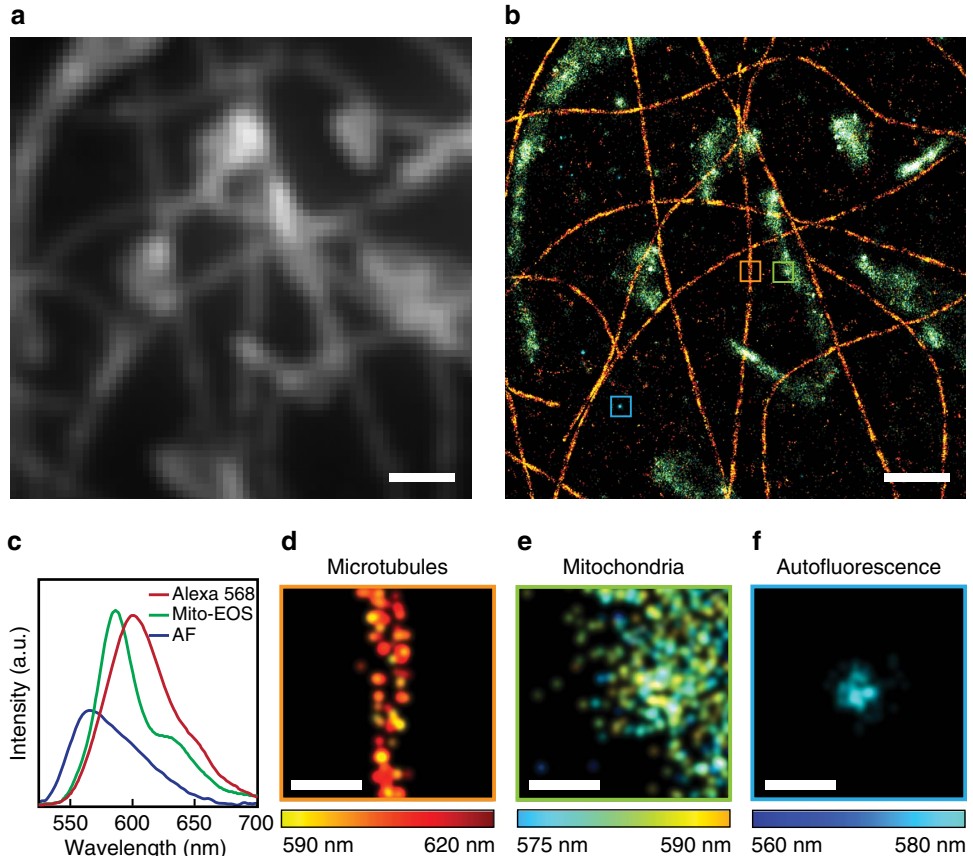

**Figure 4 | Multi-labelled SPLM imaging.** (**a**) Wide-field fluorescence image of a dual-stained COS-7 cell. (**b**) The corresponding SPLM image. Scale bars, 1 μm (**a**,**b**). (**c**) Fluorescence emission spectra of Alexa Fluor 568, Mito-EOS 4b and the distinct emission spectrum from the background. The three colours represent the spectral peak wavelengths of different molecules. Magnified views of the regions inside the coloured squares show (**d**) a single microtubule, (**e**) the edge of mitochondria and (**f**) a spot with autofluorescence emission. Scale bars, 100 nm (**d**-**f**).

different molecules. As shown in Fig. 4c, Alexa Fluor 568 has a single emission peak at 600 nm, whereas Mito-EOS 4b has a main emission peak at 580 nm and a weaker peak at 630 nm. Although the stains have similar fluorescent colours, SPLM can easily identify them due to their distinct emission spectra, which was previously challenging in reported multicolour super-resolution approaches due to limited spectral resolution[20,21]. As shown in Fig. 4d–f, magnified views of regions highlighted by the coloured squares show details from different contrasts. Supplementary Fig. 13 shows their corresponding spectra and the spectral heterogeneity.

Finally, SPLM can be used to identify artifacts from the background. As shown in Fig. 4b, we discovered scattered localization events that feature distinct emission spectra other than that of the exogenous dye molecules. These most likely from endogenous autofluorescence or from unknown sources of fluorescence induced by the use of fixatives or DNA transfection reagents[29,30]. This phenomena is overlooked by conventional PLM and may be blamed illegitimately on unspecific antibody binding[31]; however, our SPLM method provides the capabilities to reveal the potential imaging artefacts and develop a deeper understanding of their origins.

## Discussion

Use of the emission spectrum to discern the labels of fluorescent dye molecules constitutes a methodological advancement over the sequential recording used in previous multicolour experiments. Simultaneously characterizing multiple dye molecules with their spectroscopic information largely extends the combination of

discernable markers and improves imaging speed in multi-stained samples. It also provides the capability to discern imaging artefacts originated from autofluorescence through their distinct emission spectra. In addition, the demonstrated ability to distinguish the minor difference in the fluorescent spectra allows the identification of individual molecules, even among the same type of molecules. By using the spectral regression algorithm in this way, we can achieve higher spatial resolution through better use of the photons emitted by individual emitters. Moreover, image acquisition speed can be further improved by balancing the image signal-to-noise ratio (SNR) and the spectral resolution. In practical applications, high spectral resolution may not be required for identifying the vast majority of fluorescent molecules. Using the lower groove density of the grating or the shorter monochromator focal length can improve the SNR, since the available photons from each single-molecule emission occupies fewer pixels in the spectral image. This also reduces spectral overlapping and, thus, increases throughput, namely, the number of spectra that can be distinguished in one frame. Overall, the image recording can be accelerated sequentially to achieve the desired temporal resolution for particular applications[32].

Despite the success of electron microscopy and scanning probe microscope techniques, there remains a need for an optical imaging method that can uncover not only nanoscopic structures, but also the physical and chemical phenomena occurring on the nanoscale level[3–6]. We envision that SPLM can identify probes that are sensitive to properties of the nanoenvironment, which include, among many others, local pH, temperature, rotational

mobility and proximity to other probes. Thus, SPLM, which combines spectroscopy and super-resolution optical microscopy, may offer fundamentally new capabilities in many disciplines, from materials science to the life sciences.

## Methods

**Optical set-up.** The excitation source was a 532-nm diode-pumped solid-state laser with 300-mW maximum output. After passing through a laser clean-up filter (LL01-532-12.5, Semrock) and further attenuated by a set of ND filters, it was coupled to an inverted microscope body (Nikon, Eclipse Ti-U), reflected off a dichroic beam splitter (LPD02-532RU-25, Semrock), and introduced to the sample through the back focal plane of a Nikon CFI apochromat TIRF objective lens ($\times$ 100, 1.49 numerical aperture). By shifting the laser beam towards the edge of the TIRF objective with a translation stage, the emerging light reached the sample at near-critical angle of the glass–water interface, thereby illuminating only the fluorophores within a controlled range (usually a micrometer) above the coverslip surface. A 532-nm notch filter ($OD > 6$, NF01-532U-25, Semrock) was placed at the emission port to reject the reflected laser beam. The fluorescence image was coupled into a Czerny–Turner-type monochromator (SP2150, Princeton Instruments) featuring a blazed dispersive grating (150 grooves per mm). The image further divided into a non-dispersed zero-order image and a spectrally dispersed first-order spectral image. By reflecting the zero-order image back to the output port with a silver mirror, both the zero-order and the first-order images can be collected by an EMCCD camera (proEM, Princeton Instruments) simultaneously. We measured the system's spectral resolution using a Mercury–Argon calibration lamp (SPL-HGAR, Photonics Technologies) with an entrance slit width of 10 $\mu$m, as shown in Supplementary Fig. 3. By measuring the full-width half maximum of the 546.08-nm peak, the spectral resolution was calculated to be 0.63 nm. This indicates the system's spectral resolution is limited by the pixel size of the camera being used.

**SPLM imaging procedure.** The samples were placed on the microscope stage and imaged under a TIRF objective (Nikon CFI apochromat $\times$ 100, 1.49 numerical aperture). The 532-nm laser was used to excite fluorescence from Rhodamine, Alexa Fluor 568, and Mito-EOS 4b. Before acquiring SPLM images, we used relatively low-intensity 532-nm light ($\sim 0.05$ W cm$^{-2}$) to illuminate the sample and we recorded the conventional fluorescence image before switching a substantial fraction of the dye molecules to 'off' states. We then increased the 532-nm light intensity (to $\sim 2$ kW cm$^{-2}$) to rapidly switch off the dyes for SPLM imaging. The 405-nm laser was used to reactivate the fluorophores from the dark state back to the emitting state. The power of the 405-nm laser was adjusted to 0.5 W cm$^{-2}$ to maintain an appropriate fraction of the emitting fluorophores. The EMCCD camera acquired images from the monochromator at a frame rate of 50 Hz with field of view of $10 \times 10 \mu$m$^2$. Unless specifically noted, 10,000 frames were recorded to generate the super-resolution spectroscopic image.

**Imaging buffer.** A standard imaging buffer was freshly made and added to the sample prior to imaging. It contained TN buffer (50 mM Tris (pH 8.0) and 10 mM NaCl), an oxygen scavenging system (0.5 mg ml$^{-1}$ glucose oxidase (Sigma-Aldrich)), 40 $\mu$g ml$^{-1}$ catalase (Sigma-Aldrich) and 10% (w/v) glucose (Sigma-Aldrich), and 143 mM $\beta$ME (Sigma-Aldrich).

**Preparation of dye-labelled actin monomers.** Rabbit muscle actin (Cytoskeleton, Denver, CO) was suspended to 0.4 mg ml$^{-1}$ in general actin buffer (GAB, 5 mM Tris-HCl pH 8.0, 0.2 mM CaCl$_2$) supplemented with 0.2 mM ATP and 0.5 mM DTT, and then incubated on ice for 60 min to depolymerize actin oligomers. The solution was centrifuged in a 4 °C microfuge at 14 k r.p.m. for 15 min. Then, 100 $\mu$l of the actin solution was transferred from the supernatant to ultracentrifuge tubes. Alexa Fluor 532 Phalloidin solution and Alexa Fluor 568 Phalloidin solution (Life Technologies, 5 $\mu$g ml$^{-1}$ in PBS with 3% BSA) were added into the actin solutions, respectively. After incubating at 37 °C for 20 min, the solutions were centrifuged at 100,000g for 1 h. The top of the supernatant was transferred and diluted to 10$^{-9}$ M with GAB. For the single-target imaging sample, the solution containing Alexa Fluor 568-stained actin monomers was deposited onto poly-L-lysine-coated #1.5 coverslips and washed by capillary action with GAB supplemented with 0.2 mM ATP and 0.5 mM DTT. For the dual-colour sample, two solutions containing different stained actin monomers were first mixed and then deposited onto coverslips.

**Preparation of rhodamine-labelled microtubules.** Rhodamine-labelled microtubules were assembled *in vitro* by using lyophilized rhodamine-conjugated tubulin (Cytoskeleton, Denver, CO) incubated at 37 °C for 20 min in general tubulin buffer (GTB, 80 mM PIPES pH 6.9, 2 mM MgCl$_2$, and 0.5 mM EGTA) supplemented with 10% glycerol and 1 mM GTP (Cytoskeleton, Denver, CO) at a final concentration of 4 mg ml$^{-1}$. The microtubules were stabilized by incubating 25 $\mu$M taxol (Enzo Life Sciences, Farmingdale, NY) for an additional 5 min at 37 °C. For imaging, the microtubules were deposited onto poly-L-lysine-coated #1

coverslips and washed by capillary action with 100 $\mu$M GTB supplemented with 20 $\mu$M paclitaxel (taxol) and 1 mM GTP.

**Preparation for cellular imaging.** COS-7 cells (ATCC) were grown in DMEM (Gibco/Life Technologies) supplemented with 2 mM L-glutamine (Gibco/Life Technologies), 10% fetal bovine serum (Gibco/Life Technologies), and 1% penicillin (10,000 IU ml$^{-1}$)/streptomycin (10,000 $\mu$g ml$^{-1}$) (Gibco/Life Technologies) at 37 °C with 5% CO$_2$. The cells were transiently transfected with mEOS 4b-Tomm20 (Michael Davidson) using Bio-Rad Gene Pulser XCell, and were plated on 18-mm diameter #1.5 glass coverslips. After 48 h, the cells were fixed in 0.8% formaldehyde and 0.1% gluteraldehyde in PBS for 5 min at room temperature, reduced with 1% sodium borohydride for 7 min, and then further reduced in 1 mM lysine. Followed by extraction in 0.2% tween-20 in PBS for 5 min, the cells were rinsed with PBS and incubated in a blocking buffer (3% BSA (Sigma) and 1% normal goat serum in PBS) for 30 min at room temperature. The buffer was aspirated and the cells were incubated with mouse anti-$\alpha$-tubulin antibody (Sigma, 1:1,000 dilution in PBS, 3% BSA) at 37 °C for 20 min. The cells were soaked in PBS five times in 10-min intervals to rinse off the primary antibody solution. Goat-anti mouse Alexa Fluor 568 solution (Life Technologies, 5 $\mu$g ml$^{-1}$ in PBS with 3% BSA) was added to the coverslips and the cells were incubated at 37 °C for 20 min. Afterwards, the samples were rinsed in PBS for 1 h (6–7 changes) and stored in PBS at 4 °C until imaging. Prior to imaging, the sample was briefly washed once with PBS and then immediately mounted for SPLM imaging. Imaging buffer ($\sim 4 \mu$l) was dropped at the centre of a freshly cleaned glass slide, and the sample on the coverslip was mounted on the glass slide and sealed with dental cement.

**Data availability.** The data that support the findings of this study are available from the corresponding author upon request.

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

## Acknowledgements

We acknowledge the generous financial support from the National Institutes of Health (T32GM008152, T32HL076139, R01EY019951, R24EY022883 and R01CA165309), the National Science Foundation (DBI-1353952, CBET-1055379, CBET-1066776, EFRI-1240416 and EEC-1530734), and a Research Catalyst Award by Northwestern McCormick School of Engineering.

## Author contributions

B.D. designed the system, performed experiments and analysed image data. B.E.U. and L.A. prepared the samples for system characterization. S.K. and T.L.C. prepared the multi-stained cellular samples. The idea originated from the discussion between B.D. and T.Q.N. V.B., C.S. and H.F.Z. directed and supervised the project. All authors discussed the results and contributed to the manuscript.

## Additional information

**Competing financial interests:** H.F.Z. and C.S. have financial interests in Opticent Health. V.B. has financial interests in NanoCytomics. Neither of these companies funded this work. All remaining authors declare no competing financial interests.

