## [Peer review file · Nature Communications]

Reviewer #1 (Remarks to the Author):

This manuscript reports a novel optical design for spectroscopic photon localization microscopy. The key idea is to record both the spatial locations and spectral signature (using a dispersive element) at the same time. The authors demonstrate an impressive spectral resolution of 0.63 nm and a lateral resolution of 10 nm. The authors also demonstrate the system performance by imaging multi-color microtubules and mitochondria in COS-7 cells; the autofluorescence signal is identified through the spectral signature.

Overall, this is a well-written and well-organized manuscript on the spectroscopic super-resolution microscopy. There is some novelty on the optical design for recording the spatial and spectral information at the same time. However, my major concern is that, similar idea has been reported by Ke Xu's group at Berkeley (see ref 23: Ultrahigh-throughput single-molecule spectroscopy and spectrally resolved super-resolution microscopy. Nat Meth 12, 935-938 (2015)). In that Nat method paper, both 3D super-resolution and spectral imaging capabilities have been demonstrated. I do want to give credits to the authors on their improvement on spectral resolution. However, such an improvement is not particularly intriguing, as one can simply change magnification factor to improve the spectral resolution in the Nat Method setup.

At this point, I am neutral on recommending its publication in Nat Comm. The authors may need to better justify its novelty and the advantages over the previous demonstration.

Reviewer #2 (Remarks to the Author):

The manuscript claims a method for improved superresolution imaging using spectroscopic data. In addition to collecting the localization data, common in PALM/STORM experiments, in this method spectroscopic data is also recorded via a typical imaging spectrometer. The main claim of this study is that, using the spectroscopic information, the localization resolution is improved significantly (factor of 2 or better).

I find the experimental setup, which combines TIRF microscopy with simultaneous imaging of the zeroth and first order of diffraction orders interesting. The results presented in Figs. 2b and 4b seem impressive. However, it is difficult to judge the importance of these results, since it is not clear how exactly these data were obtained. There are many questions that arise while reading this paper. The main concerns I have are as follows:

1. There is no clear explanation on how spectroscopic data results in better localization, especially when these data are obtained in exchange for total loss of spatial resolution along one dimension. The supplemental material presents a series of forward simulations, but no inverse solution. In other words, the "mixing" is presented, but no procedure for "unmixing". The second and third equations in the Supplemental introduce a spectral shift, λ_i , that appears in the coordinate domain, x_i , which is likely wrong. The individual spectra on the RHS should depend on both λ and coordinate x and the spectral shift should only occur in λ . This aspect raises big questions about the method reported here.
2. Since there is no way of validating the results on microtubules and mitochondria, the authors should present control experiments on known samples.
3. Are the zeroth and first order images simultaneously in focus? The free-space propagation itself is wavelength-dependent, introduces spectral changes, which ultimately affects the measured signals.
4. It is not clear whether the spectral measurement helps with the localization only in 1D.
5. How did the authors insure that no more than one molecule emits within a diffraction spot?
6. Has any form of deconvolution been applied?

Reviewer #3 (Remarks to the Author):

Review report: Super-resolution spectroscopic microscopy via photon localization

The manuscript under review proposes a new optical method to measure high-resolution spectral information of each single molecule while taking data for normal fluorescent localization microscopy. The idea of measuring the spectral information simultaneously with the localization microscopy has been proposed in [20] to get a colored super-resolution reconstruction without sequential excitation-activation processes for different colors. However, the previous method used only the intensity ratio between two color channels to estimate the central wavelength at each pixel so the spectral resolution is not good enough in some situation [21]. A recent work uses prism to spread out the spectral information of a single molecule in the second channel to get improved spectral resolution [23]. In this paper, half of the sensor area is used to record the higher-resolution spectral information created from a blazed grating. With the additional high-resolution spectral information, the authors apply spectral un-mixing algorithm with the localization algorithm to assign each molecule with its own spectrum and achieve a claimed better spatial resolution compared to normal localization result. In addition, this technique can also get a multi-color reconstruction because of this spectral information.

Overall, the original idea of localization microscopy plus spectroscopy has been published in [20]. The proposed optical setup is new but the concept is similar to [23]. The reconstruction algorithm is novel. The contents will be interesting to the super-resolution microscopy community. The result looks better than the normal localization microscopy but there is some confusion to be addressed. I am not sure about the biological contribution made in this paper. Here are some detailed comments and questions:

1. In the normal spectral un-mixing algorithm, people set up an optimization problem to estimate the weights for each type of the molecules. Here it seems that the authors also estimate the shift of the spectrum and number of different types of molecules for a measurement at a single time frame. It is not clearly described on how they conduct this estimation. It would be great that authors can give more details on this customized spectral un-mixing algorithm.
2. For comparison of the spatial resolution, the authors use Nyquist resolution criterion proposed by [26]. This criterion says the density of the labels in a cluster determines the minimal resolvable features in that cluster. It is easy to understand how authors calculate the Nyquist resolution for PLM result. However, it is not clear how authors calculate the Nyquist resolution for SPLM result. Do you calculate the resolution based on the color-PLM result shown in Fig. 1h? If so, what is the resolution difference between color-PLM and SPLM? It seems that they share the same spatial resolution but SPLM is the average result from color-PLM.
3. Following the above point, if SPLM is the average result from color-PLM, how to cluster and merge close-by labels with similar spectrum becomes important. For dilute fluorescent beads such as Fig. 2, there may be no problem to separate each cluster. However, to separate clusters in Fig. 3 and Fig. 4 would be much harder. It would be great that authors can talk about what method or algorithm they use to do the clustering for SPLM reconstruction.
4. In the supplementary information, the authors mention that the probability of overlapping of multiple spectrums in one row is low by showing the statistics in Fig. S6. What is the corresponding experiment for this figure? Is this a general result independent of specific experiments? In addition, when we need to resolve even smaller features, higher labeling density of the fluorophores is required. At that time, the overlapping events may become more. How would the spectral un-mixing algorithm works with multiple overlapping of spectrums?

5. Other minor points:

5.1 In the 7-th line of page 6, "we were able to improve the localization precision to sub-10 nm," I guess here you want to say "Nyquist resolution criterion" instead of localization precision. The localization precision is defined as $(\text{PSF size})/\sqrt{N}$, which is different from the Nyquist resolution criterion proposed in [26].

5.2 At the end of page 6, authors should mention SD stands for standard deviation somewhere.

5.3 In Fig. S4, it would be nice to see the overall efficiency by considering all these factors.

Point-by-point response to the referees

Blue highlighted sections are our responses to reviewers' comments.

Red highlighted sections are accompanying changes made in the manuscript.

Reviewer #1:

This manuscript reports a novel optical design for spectroscopic photon localization microscopy. The key idea is to record both the spatial locations and spectral signature (using a dispersive element) at the same time. The authors demonstrate an impressive spectral resolution of 0.63 nm and a lateral resolution of 10 nm. The authors also demonstrate the system performance by imaging multi-color microtubules and mitochondria in COS-7 cells; the autofluorescence signal is identified through the spectral signature.

Overall, this is a well-written and well-organized manuscript on the spectroscopic super-resolution microscopy. There is some novelty on the optical design for recording the spatial and spectral information at the same time. However, my major concern is that, similar idea has been reported by Ke Xu's group at Berkeley (see ref 23: Ultrahigh-throughput single-molecule spectroscopy and spectrally resolved super-resolution microscopy. Nat Meth 12, 935-938 (2015)). In that Nat method paper, both 3D super-resolution and spectral imaging capabilities have been demonstrated. I do want to give credits to the authors on their improvement on spectral resolution. However, such an improvement is not particularly intriguing, as one can simply change magnification factor to improve the spectral resolution in the Nat Method setup.

At this point, I am neutral on recommending its publication in Nat Comm. The authors may need to better justify its novelty and the advantages over the previous demonstration.

Reply: We thank the reviewer for the positive comments on our manuscript. We have carefully revised the manuscript to better differentiate the focus of this paper from the study being reported by the Xu group (ref. 23). We also summarized the novelty and advantages of our method below.

While the Xu group's work mainly focuses on enabling multi-color imaging by capturing the spectroscopic information, the main innovation of our work is that SPLM significantly extends the fundamental resolution limit of photon localization microscopy by molecule discrimination through simultaneously detecting both emitted photons and their associated spectroscopic signatures. As a result, the origins of photon emissions from different single molecules can be identified according to their spectral difference, even when the molecules are within close proximity. Taking advantage of this unique capability, we demonstrated improvement in spatial (not just spectral) resolution of PALM/STORM by at least a factor of four by distinguishing the spectral variation between individual fluorophores. This method can be readily adopted by other research groups in greatly enhancing the optical resolution of photon localization microscopy without the need to modify their existing staining methods and protocols. This new resolving capability can potentially provide new insights into biological phenomena and enable significant research progress to be made in the life sciences.

To distinguish minute differences in fluorescence spectra and to achieve a high molecular specificity in spatial resolution, sub-nanometer spectral resolution is a prerequisite. We also developed algorithms for SPLM to overcome the new challenges that emerged from the improved spectral resolution, including background subtraction and spectral overlap unmixing. In addition, we established and tested a spectral regression method to improve spatial resolution, in which the criteria are setup on the experimental single molecule spectroscopy data acquired by SPLM simultaneously.

We have emphasized the novelty in the introduction:

“While other optical configuration was recently reported in enabling multicolor super-resolution imaging in 3D²³, SPLM permits fluorescence spectral analysis of individual molecules with experimentally demonstrated sub-nanometer spectral resolution and sub-10 nanometer spatial resolution. ... By employing spectral unmixing¹¹ and regression²⁴, individual fluorescent molecules located within a close proximity can be still distinguished according to their emission spectra. Taking advantage of this unique capability, SPLM significantly extends the fundamental spatial resolution limit of photon localization microscopy by at least a factor of four through combining photons from the same fluorophore.”

We further added Supplementary Note 2 to provide a detailed explanation of the spatial resolution improvement via spectral regression.

Reviewer #2:

The manuscript claims a method for improved superresolution imaging using spectroscopic data. In addition to collecting the localization data, common in PALM/STORM experiments, in this method spectroscopic data is also recorded via a typical imaging spectrometer. The main claim of this study is that, using the spectroscopic information, the localization resolution is improved significantly (factor of 2 or better).

I find the experimental setup, which combines TIRF microscopy with simultaneous imaging of the zeroth and first order of diffraction orders interesting. The results presented in Figs. 2b and 4b seem impressive. However, it is difficult to judge the importance of these results, since it is not clear how exactly these data were obtained. There are many questions that arise while reading this paper. The main concerns I have are as follows:

Reply: We thank the reviewer for the positive evaluation. We have revised the manuscript according to reviewer's constructive comments to improve the clarity of the manuscript.

1. *There is no clear explanation on how spectroscopic data results in better localization, especially when these data are obtained in exchange for total loss of spatial resolution along one dimension. The supplemental material presents a series of forward simulations, but no inverse solution. In other words, the "mixing" is presented, but no procedure for "unmixing". The second and third equations in the Supplemental introduce a spectral shift, λ_i , that appears in the coordinate domain, x_i , which is likely wrong. The individual spectra on the RHS should depend on both λ and coordinate x and the spectral shift should only occur in λ . This aspect raises big questions about the method reported here.*

Reply: SPLM simultaneously records both spatial position and spectroscopic information. Both the zero-order and the first-order diffractions from the grating were recorded by the same EMCCD camera, as shown in Figure 1b (also cropped and attached below). The zero-order images are firstly analyzed by using the standard localization algorithm to determine the locations of individual blinking events, which is identical to the processing used in PALM/STORM. The collected centroid position serves two roles: (1) to determine the location of each activated fluorescent molecule (shown as numbered crosses in Figure 1e) and (2) to establish a reference point to obtain the corresponding emission spectrum in the first-order image (shown as numbered crosses in Figure 1b).

Cropped from Figure 1: (b) Both the zero-order and the first-order diffractions from the grating were recorded simultaneously using the same EMCCD camera. (e) The localization algorithm was used to determine the spatial locations of each blinking, illustrated by numbered crosses. These locations can be further used as the inherent reference points for spectral calibration of the emission spectra in the first-order image, shown as denoted crosses in (b). (f) Representative spectra from two individual blinking events (highlighted by the colored arrows in (e)).

As the spectrum provides critical information for molecular discrimination, photons collected from the same molecule over multiple frames can be combined to increase the total photon count used in localization analysis and, therefore, to achieve a higher precision from the localization analysis. We added Supplementary Note 2 to provide a detailed explanation of the resolution improvement via spectral regression. As shown in Supplementary Figure S11, a simulation was used to demonstrate the concept and to quantitatively evaluate the improved spatial resolution. As suggested by reviewer, we also included the “inverse” simulation, as shown in Supplementary Figure S12, to demonstrate the feasibility and performance of our spectral regression method when dealing with a densely labeled structure. Related discussion can be also found in Supplementary Note 2 as given below.

“Supplementary Note 2

Improving localization precision via spectral regression

The precision of identifying the centroid location (σ) can be approximated by the probability equation²

$$\sigma = \sqrt{\left(\frac{s_i^2}{N}\right) + \left(\frac{a^2/12}{N}\right) + \left(\frac{8\pi s_i^4 b^2}{a^2 N^2}\right)},$$

where s_i is the standard deviation of the Gaussian fit in x and y direction; N is the number of detected photons; a is the pixel size of the CCD camera; and b is the standard deviation of the CCD background. As we can see, the localization uncertainty is proportional to the inverse square root of the number of detected photons.

To further validate the resolution improvement with spectral regression, we first performed a simulation in the case of a single molecule and calculated the precision after regression, as shown in Supplementary Figure S11. In general, the regression can be judged by any molecular specific parameter, such as intensity, polarization, anisotropy, and emission spectrum. Since we

are considering a single molecule case in this simulation, all blinking events being collected are naturally from the same molecule and no regression is necessary. The feasibility of our spectral regression algorithm will be discussed in the next case of simulation using the line pattern.

In order to examine the localization precision with respect to the number of blinking (NB) from each molecule, we first generated a movie that consisted of NB frames with a pixel size of 100 nm. In each frame, a single blinking event at diffraction-limited resolution (predefined by the objective NA at wavelength of 600 nm) was superimposed on a Gaussian noise background (Supplementary Figure S11a). The total photon count and noise level were adjusted to match our experimental conditions. After reconstruction using a standard PALM/STORM algorithm, positions of all the collected localizations were plotted as white crosses. Their centroid, which represents the result of the regression, was further plotted as a red cross. As shown in Supplementary Figure S11b, we tested 500 randomly generated cases to evaluate the localization precision of centroids by plotting the histogram along one of the lateral directions (x-axis). When NB=1, the result shows the precision of conventional PLM (37.4 nm) since no regression was used. As NB increases, the localization precision improves, reaching a resolution to 5.1 nm (7.3-fold improvement) when NB=50. In practice, the NB acquired from experiments normally ranges between 10 and 20. Thus the experimental resolution we can typically achieve is between 7.9 nm and 11.7 nm. To achieve even higher resolution, we can prolong image acquisition time and acquire more blinking events from the same molecules if they still generate blinking. These results illustrate the principle of using spectral regression to improve image resolution in SPLM.

Supplementary Figure S11. (a) Ground truth of the molecule (upper panel) and a simulated single blinking event at diffraction-limited resolution (lower panel). (b) Improved resolution using molecular regression. White crosses denoted positions of all localizations in each test and red crosses denoted their centroids. Localization precision with respect to the number of blinking

(NB) is calculated from the FWHM of the histogram, which are 37.4 nm, 16.5 nm, 11.7 nm, 7.9 nm and 5.1 nm for NB=1, 5, 10, 20 and 50, respectively.

Since spectroscopic information provides one of the most important evidence for molecular discrimination, we hereby used SPLM to realize spectral regression to improve spatial resolution. To demonstrate the feasibility and the performance of our spectral regression method for densely labeled molecules, we performed a simulation using a straight-line pattern consisting of randomly arranged molecules with line density of 1 nm^{-1} . Supplementary Figure S12a shows the ground truth of the line pattern. To generate the simulated movie for spectral regression of a single molecule type, the emission spectrum of each molecule was modeled after the emission properties of Alexa Fluor 568, where the peak emission is 610 nm. A fixed wavelength shift in the range of ± 11 nm was further assigned to the modeled spectrum, which represents the spectral variation of Alexa Fluor 568 observed in the single molecule experiment shown in Supplementary Figure S10. In the generated movie, each frame contains one stochastic blinking event in the zeroth order and a spectrum in the first order images, which are similar to the simulated image shown in Supplementary Figure S7. All simulation conditions, including sizes of the pixel and the point spread function, photon number and background noise level, were set to be exactly the same as the previous simulation for a fair comparison. We then performed reconstruction and collected locations of each blinking event and the associated spectra. The spectral shift of each spectrum $\Delta\lambda_i$ was calculated based on the shift of the spectral centroid by using the following equation:

$$\Delta\lambda_i = \frac{\sum_{\lambda} \lambda I(\lambda)}{\sum_{\lambda} I(\lambda)} - \Delta\lambda_0,$$

where λ is the wavelength; $I(\lambda)$ represents the spectral intensity at λ ; and $\Delta\lambda_0$ is the centroid of the un-shifted spectrum.

To perform the appropriate spectral regression, we have to establish reasonable criteria. First, only localizations within the range of localization precision in the zeroth-order image (37.4 nm) were considered. The criteria in the spectral domain were derived from the experimental single molecule spectroscopy data acquired by SPLM simultaneously, as shown in Figure 3e. The spectral shift from the same molecule should be less than 2 nm, which was derived from the maximum standard deviation of spectral variation from the single molecule experiment (Supplementary Figure S10). Based on experimental observation, an acceptable variation of intensity was set to be $\pm 10\%$ of the integrated intensity of the spectrum. All localizations were then judged by these criteria to be clustered and merged as molecules.

Supplementary Figure S12b shows the reconstructed images of PLM and SPLM, where only SPLM uses the spectral regression as mentioned above. Resolution improvement can be clearly observed when NB increases. This is consistent with the previous simulation performed in a single molecule situation. Additionally, we calculated the resulting resolution of SPLM as a function of NB by averaging images along the vertical direction (y-axis), as shown in Supplementary Figure S12c. In conventional PLM without regression, the resolution obtained after reconstruction is independent of the NB, remaining identical as 38.1 ± 4.1 nm. In Supplementary Figure S12d, the resolutions of SPLM are respectively improved to be 17.8 ± 1.7 nm, 12.3 ± 1.4 nm, 8.3 ± 1.0 nm and 5.3 ± 0.8 nm when NBs from each molecule are 1, 5, 10, 20 and 50. They follow the trend of P_0/\sqrt{NB} , where P_0 is the localization precision when NB=1, as

shown in Supplementary Figure S12e. This trend suggests that spectral regression indeed improves the localization precision by increasing the photon counts for localization analysis. Even with the strict rejection criteria described above, it's worth mentioning that $90\pm 5\%$ and $85\pm 6\%$ of localizations for the individual molecules can be recovered when $NB=10$ and $NB=20$, respectively, when compared with the ground truth. This demonstrates the feasibility of spectral regression to be performed in densely labeled samples and it sets the foundation of image reconstruction in Figure 3 and 4.

Supplementary Figure S12. (a) Ground truth of a random generated line pattern. The pattern was used to simulate a movie consisting of both zero-order and first order images for testing the performance of spectral regression method. The number of blinking (NB) from each molecule is set to be 1, 5, 10, 20 and 50, respectively. (b) PLM and SPLM images reconstructed from the simulated movie. Spectral regression was performed in SPLM reconstruction. Localization precision was calculated by average images along y-axis and was then normalized for comparison. (c) PLM images show same localization precision regardless the NB. (d) SPLM images have improved localization precision when NB increases. (e) Localization precision with respect to NB in SPLM. Black line is the curve calculated by P_0/\sqrt{NB} , where P_0 is the localization precision at $NB=1$.

For multi-labeled sample, we first classified localizations by molecule types based on their spectra and then applied separate criteria in spectral regression for molecules of different types based on the analysis of data shown in Supplementary Figure S13.

We have further revised the manuscript in addressing the reviewer's confusion in regards the definition of the spectral shift (λ_i). Since we used the centroid position collected in zero-order image to establish a reference point to the corresponding emission spectrum measured from the first-order image (shown as numbered crosses in Figure 1b), The spectral unmixing was

performed using the pixel coordinate before converting the spectra to the wavelength coordinate (Fig. S5). While x_i indicates the inherent reference points for spectral calibration of the emission spectra, λ_i caused by the spectral shift from the conformational heterogeneity of the fluorophore is previously defined in the pixel coordinate. Since the system dispersion is nearly linear, λ_i can be calculated as the shifted wavelength divided by the spectral resolution (e.g. 10 nm/0.63 nm=16 pixels). However, the symbol we chose to represent the shift looks more like a wavelength shift and caused this misunderstanding. Therefore we revised the symbol of spectral shift to $\Delta\lambda_i$ and defined the corresponding pixel shift as d_i .

To avoid further confusion, we have revised the discussion in the Supplementary Note 1 as follows:

“For single molecule spectroscopy, we also have to consider the spectral shift $\Delta\lambda_i$ from the underlying conformational heterogeneity. The observed spectrum S can be further expressed as

$$S = \sum_{i=1}^n a_i s(x_i + d_i) + w,$$

where d_i is corresponding shift in pixel domain, which can be calculated as $\Delta\lambda_i/0.63\text{nm}$.”

2. *Since there is no way of validating the results on microtubules and mitochondria, the authors should present control experiments on known samples.*

Reply: Due to biological complexity, it’s challenging to find a completely known biology sample for control experiments at sub-10-nm length scale. We chose the rhodamine-labeled microtubules sample for validation because it is commercially available¹ (Cytoskeleton, Denver, CO), synthesized *in vitro*, and the structure of these microtubules has been well examined and published elsewhere²⁻⁵, which makes it a widely accepted nanoscale biological sample for validating super resolution imaging methods⁶⁻⁸. A single microtubule in mammalian cells usually consists of 13 protofilaments to form a 24-nm wide hollow cylinder³⁻⁵. To further clarify the structure of the microtubules we fabricated, we included a TEM image of a microtubule and the line profile in Figure R1. The profile of the cross section shown in TEM image agrees with that shown in Figure 3d.

Figure R1. (a) TEM image of ex vivo synthesized microtubules sample. (b) Schematic of the

microtubule cross section. (c) TEM cross-sectional profile from the highlighted position shows the hollow structure of the microtubule, which agrees with our SPLM image.

As we mentioned previously, to demonstrate the feasibility of our method we also performed a simulation using a known structure to validate the resolution improvement with respect to numbers of blinking from each molecule (Supplementary Figure S12).

1. <http://www.cytoskeleton.com/tl590m>
2. Conde, C. & Cáceres, A. Microtubule assembly, organization and dynamics in axons and dendrites. *Nature Reviews Neuroscience* **10**, 319-332 (2009)
3. Desai, A. & Mitchison, T. Microtubule polymerization dynamics. *Annu. Rev. Cell Dev. Biol.* **13**, 83–117 (1997).
4. Nogales, E., Wolf, S. & Downing, K. H. Structure of the $\alpha\beta$ tubulin dimer by electron crystallography. *Nature* **391**, 199–203 (1998).
5. Lowe, J., Li, H. & Nogales, E. Refined structure of a β -tubulin at 3.5 Å resolution. *J. Mol. Biol.* **313**, 1045–1057 (2001).
6. Dempsey, G. T., Vaughan, J. C., Chen, K. H., Bates, M. & Zhuang, X. W. Evaluation of fluorophores for optimal performance in localization-based super-resolution imaging. *Nat. Methods* **8**, 1027-1036 (2011).
7. Jia, S., Vaughan, J. C. & Zhuang, X. W. Isotropic three-dimensional super-resolution imaging with a self-bending point spread function. *Nat. Photonics* **8**, 302-306 (2014).
8. Kiuchi, T., Higuchi, M., Takamura, A., Maruoka, M. & Watanabe, N. Multitarget super-resolution microscopy with high-density labeling by exchangeable probes. *Nat. Methods* **12**, 893-893 (2015).

3. *Are the zeroth and first order images simultaneously in focus? The free-space propagation itself is wavelength-dependent, introduces spectral changes, which ultimately affects the measured signals.*

Reply: A chromatically correct imaging spectrometer (SP2150, Princeton Instruments) was used to collect both zero-order and first-order images. The hyperbolic mirror in the spectrometer is designed to ensure both zero-order and first-order images are simultaneously in focus. Additionally, the system has been carefully aligned to ensure the best focus of both zero-order and first-order images and demonstrated by the spectrum measured from a Mercury-Argon calibration lamp using our imaging spectrometer (Figure S3). We agree the free-space propagation is wavelength-dependent and may introduce spectral changes. Since the apochromat TIRF objective lens (Nikon) provides chromatic correction from 435 nm to 1064 nm and the commercial microscope (Nikon Ti-U) used in our experiments are also chromatically corrected and thus, the spectral change induced by the optical path has been minimized.

4. *It is not clear whether the spectral measurement helps with the localization only in 1D.*

Reply: The improvement in spatial resolution is experimentally demonstrated in 2D. We have added detailed discussion in “Supplementary Note 2” to explain how spectral regression can help improve the spatial resolution.

To clarify, the spectral measurement doesn't directly contribute to the resolution improvement. Instead, spectral measurement provides critical information for molecular discrimination. With this information, we can combine multiple blinking events emitted from the same molecule captured from different frames to achieve a higher localization precision as a result of increased photon counts, as shown in Fig. S11.

5. *How did the authors insure that no more than one molecule emits within a diffraction spot?*

Reply: As the foundation of any single molecule localization microscopy, most fluorescent molecules were turned to their dark state. Only small population of fluorescent molecules was allowed to emit simultaneously. As the results, the probability of more than one molecule emitting simultaneously within a diffraction-limited spot is rather low. In the rare events the multiple emissions occur within the diffraction limit spot, the PSF will exhibit elliptical shape. These localizations can be excluded by the reconstruction algorithm by analyzing the ellipticity and will not be used in the spectral analysis eliminating the potential artifacts in the reconstructed image.

6. *Has any form of deconvolution been applied?*

Reply: We didn't apply any form of deconvolution in our reconstruction process.

Reviewer #3:

Review report: Super-resolution spectroscopic microscopy via photon localization

The manuscript under review proposes a new optical method to measure high-resolution spectral information of each single molecule while taking data for normal fluorescent localization microscopy. The idea of measuring the spectral information simultaneously with the localization microscopy has been proposed in [20] to get a colored super-resolution reconstruction without sequential excitation-activation processes for different colors. However, the previous method used only the intensity ratio between two color channels to estimate the central wavelength at each pixel so the spectral resolution is not good enough in some situation [21]. A recent work uses prism to spread out the spectral information of a single molecule in the second channel to get improved spectral resolution [23]. In this paper, half of the sensor area is used to record the higher-resolution spectral information created from a blazed grating. With the additional high-resolution spectral information, the authors apply spectral un-mixing algorithm with the localization algorithm to assign each molecule with its own spectrum and achieve a claimed better spatial resolution compared to normal localization result. In addition, this technique can also get a multi-color reconstruction because of this spectral information.

Overall, the original idea of localization microscopy plus spectroscopy has been published in [20]. The proposed optical setup is new but the concept is similar to [23]. The reconstruction algorithm is novel. The contents will be interesting to the super-resolution microscopy community. The result looks better than the normal localization microscopy but there is some confusion to be addressed. I am not sure about the biological contribution made in this paper.

Reply: We thank the reviewer for the careful evaluation and positive comments on our manuscript.

As we mentioned in the reply to Reviewer #1, although the idea of photon localization microscopy plus spectroscopy was previously reported (ref. 23), we are the first to report the concept and the experimental demonstration of using the spectral variation between individual fluorophores to improve the spatial resolution by at least a factor of four in super-resolution imaging. This method can be readily adopted by other research group in greatly enhancing the optical resolution of the localization microscopy without the need to modify the existing staining method and protocols. This new resolving capability can potentially provide new insights into biological phenomena and enable significant research progress to be made in the life sciences.

More importantly, such improvement can be only realized by using SPLM, which has a sub-nanometer spectral resolution to distinguish minor spectral shifts from the underlying conformational heterogeneity of different molecules. We have also developed algorithms for SPLM to overcome new problems challenges that encountered by the improved spectral resolution, e.g. subtracting the scattering background and unmixing overlapping spectra. In addition, we established and tested a spectral regression method to improve spatial resolution, in which the criteria are setup on the experimental single molecule spectroscopy data acquired by SPLM simultaneously.

Here are some detailed comments and questions:

1. *In the normal spectral un-mixing algorithm, people set up an optimization problem to estimate the weights for each type of the molecules. Here it seems that the authors also estimate the shift of the spectrum and number of different types of molecules for a measurement at a single time frame. It is not clearly described on how they conduct this estimation. It would be great that authors can give more details on this customized spectral un-mixing algorithm.*

Reply: As modified from a linear spectral unmixing algorithm, it's indeed an optimization problem with utilizing the molecule type and the spectral shift as free parameters. In the current manuscript, we applied a linear least-squares solver, a built-in solver in MATLAB, for a proof-of-concept demonstration. We are currently exploring better algorithms, which can provide higher efficiency, can handle highly overlapped spectra, and can potentially perform blind spectral unmixing without prior knowledge of fluorophores' emission spectra. We are currently developing an open-source software based on ImageJ to serve the community who are interested in using SPLM in their research. We intend to publish the new software in the near future when it is ready.

We revised Supplementary Note 1 to mention the solver we used in our modified spectral unmixing algorithm.

Line 12 in Supplementary Note 1:

“As an optimization process, we applied a linear least-squares solver, a built-in solver in MATLAB, to solve the above equation with a known number of localization events, where a_i and x_i are free parameters. After conducting unmixing, overlapped spectra can be separated as shown in Supplementary Figure S5.”

2. *For comparison of the spatial resolution, the authors use Nyquist resolution criterion proposed by [26]. This criterion says the density of the labels in a cluster determines the minimal resolvable features in that cluster. It is easy to understand how authors calculate the Nyquist resolution for PLM result. However, it is not clear how authors calculate the Nyquist resolution for SPLM result. Do you calculate the resolution based on the color-PLM result shown in Fig. 1h? If so, what is the resolution difference between color-PLM and SPLM? It seems that they share the same spatial resolution but SPLM is the average result from color-PLM.*

Reply: We apologize for the confusion resulted from using terminologies of Nyquist resolution criterion and localization precision. We agree with the reviewer that the ultimate resolution depends on both Nyquist resolution criterion and localization precision. In single molecule imaging, the resolution improvement between color-PLM and SPLM should be compared by their localization precision rather than the Nyquist resolution. We have corrected all terminologies in the revised manuscript accordingly and added a clear definition of the localization precision in Supplementary Note 2.

“The precision of identifying the centroid location (σ) can be approximated by the probability equation²

$$\sigma = \sqrt{\left(\frac{s_i^2}{N}\right) + \left(\frac{a^2/12}{N}\right) + \left(\frac{8\pi s_i^4 b^2}{a^2 N^2}\right)},$$

where s_i is the standard deviation of the Gaussian fit in x and y direction; N is the number of detected photons; a is the pixel size of the CCD camera; and b is the standard deviation of the CCD background. As we can see, the localization uncertainty is proportional to the inverse square root of the number of detected photons.”

In color-PLM, the resolution is calculated from the photons of each individual blinking event, which is essentially the same as the resolution of non-colored PLM. In SPLM, the resolution is calculated by accumulating all the photons emitted from the same molecule to generate a higher localization precision and, thus, higher spatial resolution.

To fully address the reviewer’s question, we added a discussion in Supplementary Note 2 and a simulation as shown in Supplementary Figure S11 to further explain the resolution improvement via molecular spectral regression.

3. *Following the above point, if SPLM is the average result from color-PLM, how to cluster and merge close-by labels with similar spectrum becomes important. For dilute fluorescent beads such as Fig. 2, there may be no problem to separate each cluster. However, to separate clusters in Fig. 3 and Fig. 4 would be much harder. It would be great that authors can talk about what method or algorithm they use to do the clustering for SPLM reconstruction.*

Reply: We thank the reviewer for pointing out this important aspect. The method used for regression is rather straightforward. As we demonstrated in Supplementary Figure S10, each individual molecule has similar emission spectra and photon counts in stochastic blinking events, while different molecules exhibit clear spectral variations. We therefore set criteria by simultaneously judging the blinking location, spectral shift, and spectral intensity to each cluster and merge the molecule’s blinking events. For the multi-labeled sample, we first classified each cluster by molecule types based on their mean spectra and then applied separated criteria during spectral regression for molecules of different types based on the analysis of data shown in Supplementary Figure S13.

We further added detailed discussions in Supplementary Note 2 and a simulation shown in Supplementary Figure S12 to explain the feasibility and performance of our spectral regression method.

“Supplementary Note 2

To further validate the resolution improvement with spectral regression, we first performed a simulation in the case of a single molecule and calculated the precision after regression, as shown in Supplementary Figure S11. In general, the regression can be judged by any molecular specific parameter, such as intensity, polarization, anisotropy, and emission spectrum. Since we are considering a single molecule case in this simulation, all blinking events being collected are

naturally from the same molecule and no regression is necessary. The feasibility of our spectral regression algorithm will be discussed in the next case of simulation using the line pattern.

In order to examine the localization precision with respect to the number of blinking (NB) from each molecule, we first generated a movie that consisted of NB frames with a pixel size of 100 nm. In each frame, a single blinking event at diffraction-limited resolution (predefined by the objective NA at wavelength of 600 nm) was superimposed on a Gaussian noise background (Supplementary Figure S11a). The total photon count and noise level were adjusted to match our experimental conditions. After reconstruction using a standard PALM/STORM algorithm, positions of all the collected localizations were plotted as white crosses. Their centroid, which represents the result of the regression, was further plotted as a red cross. As shown in Supplementary Figure S11b, we tested 500 randomly generated cases to evaluate the localization precision of centroids by plotting the histogram along one of the lateral directions (x-axis). When NB=1, the result shows the precision of conventional PLM (37.4 nm) since no regression was used. As NB increases, the localization precision improves, reaching a resolution to 5.1 nm (7.3-fold improvement) when NB=50. In practice, the NB acquired from experiments normally ranges between 10 and 20. Thus the experimental resolution we can typically achieve is between 7.9 nm and 11.7 nm. To achieve even higher resolution, we can prolong image acquisition time and acquire more blinking events from the same molecules if they still generate blinking. These results illustrate the principle of using spectral regression to improve image resolution in SPLM.

Supplementary Figure S11. (a) Ground truth of the molecule (upper panel) and a simulated single blinking event at diffraction-limited resolution (lower panel). (b) Improved resolution using molecular regression. White crosses denoted positions of all localizations in each test and red crosses denoted their centroids. Localization precision with respect to the number of blinking (NB) is calculated from the FWHM of the histogram, which are 37.4 nm, 16.5 nm, 11.7 nm, 7.9 nm and 5.1 nm for NB=1, 5, 10, 20 and 50, respectively.

Since spectroscopic information provides one of the most important evidence for molecular discrimination, we hereby used SPLM to realize spectral regression to improve spatial resolution. To demonstrate the feasibility and the performance of our spectral regression method for densely labeled molecules, we performed a simulation using a straight-line pattern consisting of randomly arranged molecules with line density of 1 nm^{-1} . Supplementary Figure S12a shows the ground truth of the line pattern. To generate the simulated movie for spectral regression of a single molecule type, the emission spectrum of each molecule was modeled after the emission properties of Alexa Fluor 568, where the peak emission is 610 nm. A fixed wavelength shift in the range of $\pm 11 \text{ nm}$ was further assigned to the modeled spectrum, which represents the spectral variation of Alexa Fluor 568 observed in the single molecule experiment shown in Supplementary Figure S10. In the generated movie, each frame contains one stochastic blinking event in the zeroth order and a spectrum in the first order images, which are similar to the simulated image shown in Supplementary Figure S7. All simulation conditions, including sizes of the pixel and the point spread function, photon number and background noise level, were set to be exactly the same as the previous simulation for a fair comparison. We then performed reconstruction and collected locations of each blinking event and the associated spectra. The spectral shift of each spectrum $\Delta\lambda_i$ was calculated based on the shift of the spectral centroid by using the following equation:

$$\Delta\lambda_i = \frac{\sum_{\lambda} \lambda I(\lambda)}{\sum_{\lambda} I(\lambda)} - \Delta\lambda_0,$$

where λ is the wavelength; $I(\lambda)$ represents the spectral intensity at λ ; and $\Delta\lambda_0$ is the centroid of the un-shifted spectrum.

To perform the appropriate spectral regression, we have to establish reasonable criteria. First, only localizations within the range of localization precision in the zeroth-order image (37.4 nm) were considered. The criteria in the spectral domain were derived from the experimental single molecule spectroscopy data acquired by SPLM simultaneously, as shown in Figure 3e. The spectral shift from the same molecule should be less than 2 nm, which was derived from the maximum standard deviation of spectral variation from the single molecule experiment (Supplementary Figure S10). Based on experimental observation, an acceptable variation of intensity was set to be $\pm 10\%$ of the integrated intensity of the spectrum. All localizations were then judged by these criteria to be clustered and merged as molecules.

Supplementary Figure S12b shows the reconstructed images of PLM and SPLM, where only SPLM uses the spectral regression as mentioned above. Resolution improvement can be clearly observed when NB increases. This is consistent with the previous simulation performed in a single molecule situation. Additionally, we calculated the resulting resolution of SPLM as a function of NB by averaging images along the vertical direction (y-axis), as shown in Supplementary Figure S12c. In conventional PLM without regression, the resolution obtained after reconstruction is independent of the NB, remaining identical as $38.1 \pm 4.1 \text{ nm}$. In Supplementary Figure S12d, the resolutions of SPLM are respectively improved to be $17.8 \pm 1.7 \text{ nm}$, $12.3 \pm 1.4 \text{ nm}$, $8.3 \pm 1.0 \text{ nm}$ and $5.3 \pm 0.8 \text{ nm}$ when NBs from each molecule are 1, 5, 10, 20 and 50. They follow the trend of P_0/\sqrt{NB} , where P_0 is the localization precision when NB=1, as shown in Supplementary Figure S12e. This trend suggests that spectral regression indeed improves the localization precision by increasing the photon counts for localization analysis.

Even with the strict rejection criteria described above, it's worth mentioning that $90\pm 5\%$ and $85\pm 6\%$ of localizations for the individual molecules can be recovered when $NB=10$ and $NB=20$, respectively, when compared with the ground truth. This demonstrates the feasibility of spectral regression to be performed in densely labeled samples and it sets the foundation of image reconstruction in Figure 3 and 4.

For multi-labeled sample, we first classified localizations by molecule types based on their spectra and then applied separate criteria in spectral regression for molecules of different types based on the analysis of data shown in Supplementary Figure S13.

Supplementary Figure S12. (a) Ground truth of a random generated line pattern. The pattern was used to simulate a movie consisting of both zero-order and first order images for testing the performance of spectral regression method. The number of blinking (NB) from each molecule is set to be 1, 5, 10, 20 and 50, respectively. (b) PLM and SPLM images reconstructed from the simulated movie. Spectral regression was performed in SPLM reconstruction. Localization precision was calculated by average images along y-axis and was then normalized for comparison. (c) PLM images show same localization precision regardless the NB. (d) SPLM images have improved localization precision when NB increases. (e) Localization precision with respect to NB in SPLM. Black line is the curve calculated by P_0/\sqrt{NB} , where P_0 is the localization precision at $NB=1$.”

4. In the supplementary information, the authors mention that the probability of overlapping of multiple spectrums in one row is low by showing the statistics in Fig. S6. What is the corresponding experiment for this figure? Is this a general result independent of specific experiments? In addition, when we need to resolve even smaller features, higher labeling density of the fluorophores is required. At that time, the overlapping events may become more. How

would the spectral un-mixing algorithm works with multiple overlapping of spectrums?

Reply: Supplementary Figure S6 is a representative statistical result from the spectral overlap from 10,000 blinking events acquired from the Rhodamine-labeled microtubule sample used in Figure 3. We added this information to the revised supplementary information. Although the rate of overlap is highly dependent on the labeling density and the feature size of the sample, it can be reduced by increasing the excitation light intensity to reduce the overall blinking density. We can therefore carefully control the imaging condition to avoid a massive number of overlaps while still maintaining a sufficient number of localizations to achieve a satisfactory image quality.

In addition, the stochastic blinking events are also separated in time, therefore massive overlap in single frames is unlikely, even for high-density labeled sample. Longer imaging time can be used to obtain satisfactory details from an extremely high-density labeled area.

5. Other minor points:

5.1 In the 7-th line of page 6, "we were able to improve the localization precision to sub-10 nm," I guess here you want to say "Nyquist resolution criterion" instead of localization precision. The localization precision is defined as $(PSF\ size)/\sqrt{N}$, which is different from the Nyquist resolution criterion proposed in [26].

Reply: As we mentioned in the reply to Comment 2, we corrected all the terminologies regarding localization precision and Nyquist criterion in the revised manuscript.

5.2 At the end of page 6, authors should mention SD stands for standard deviation somewhere.

Reply: We added the definition of SD (standard deviation) in the revised manuscript.

5.3 In Fig. S4, it would be nice to see the overall efficiency by considering all these factors.

Reply: We added the overall efficiency in Supplementary Figure S4 as shown below:

Supplementary Figure S4. Efficiency of the optical system. Blue line: transmission spectra of the combination of dichroic beamsplitter (LPD02-532RU-25, Semrock) and long pass filter (BLP01-532R-25, Semrock) in the filter cube. Green line: transmission spectra of the 532-nm notch filters (NF01-532U-25, Semrock). Red line: reflectance spectrum of aluminum mirrors. Purple line: the wavelength dependent quantum efficiency (QE) of the EMCCD with broadband anti-reflection (BBAR) coating. Orange line: the diffraction efficiency of the diffraction grating. Black line: the overall efficiency of the system. Cyan line: the overall efficiency of the first order.

Reviewer #1 (Remarks to the Author):

In the revised manuscript, the authors adequately address the comments of the reviewers. In particular, I found the new discussion in Supplementary Note 2 very useful. Therefore, I recommend its publication in Nature Communications.

Reviewer #2 (Remarks to the Author):

I appreciate the effort that the authors devoted to answering the concerns raised by me and the other reviewers. The rebuttal is long and difficult to navigate as the authors chose to copy paste multiple times the same text. But that is secondary.

I have to say that the main question, which is: "how exactly is the spectral information improving resolution?", remains unanswered in my opinion. The authors bring in now an old formula for the localization precision in PALM/STORM. This formula has no spectral dependence (although the same symbol, s , is used for the std. dev. of the Gaussian fit and the spectrum in the previous equation, which is confusing). The authors claim that they capture more photons via the spectral measurement. However, that cannot be true. Introducing the optical elements of the spectrometer surely reduces the overall number of photons detected. This has to be clarified.

Related to my previous point 4, which is asking about the fact that the spectroscopic data is traded for 1D spatial data. The answer was:

"To clarify, the spectral measurement doesn't directly contribute to the resolution improvement. Instead, spectral measurement provides critical information for molecular discrimination. "

What does this mean?

"With this information, we can combine multiple blinking events emitted from the same molecule captured from different frames to achieve a higher localization precision as a result of increased photon counts, as shown in Fig. S11."

Again, I do not see the point about the photon count here.

The simulation in Fig. 7 hints that if two fluorophores are perfectly aligned in the direction of the spectral spread, then their localization can be decoupled. How about localization in the other direction?

Also, it just occurred to me at the last reading that the authors define the spectral resolution purely through pixel sampling, giving 0.63nm. Resolution and sampling are different quantities, because the spectral resolution is also affected by diffraction, it must depend on the properties of the grating (number of grooves illuminated), the distance to detector, etc. The resolution should be recalculated or measured.

Reviewer #3 (Remarks to the Author):

[The referee believes the work is now acceptable for publication, with no further comments for the authors]

Point-by-point response to the referees

Reviewer #1:

In the revised manuscript, the authors adequately address the comments of the reviewers. In particular, I found the new discussion in Supplementary Note 2 very useful. Therefore, I recommend its publication in Nature Communications.

Reply: We greatly appreciate the reviewer's efforts in carefully reviewing both the original and revised manuscripts and the valuable suggestions.

Reviewer #2:

I appreciate the effort that the authors devoted to answering the concerns raised by me and the other reviewers. The rebuttal is long and difficult to navigate as the authors chose to copy paste multiple times the same text. But that is secondary.

1, I have to say that the main question, which is: "how exactly is the spectral information improving resolution?", remains unanswered in my opinion. The authors bring in now an old formula for the localization precision in PALM/STORM. This formula has no spectral dependence (although the same symbol, s , is used for the std. dev. of the Gaussian fit and the spectrum in the previous equation, which is confusing). The authors claim that they capture more photons via the spectral measurement. However, that cannot be true. Introducing the optical elements of the spectrometer surely reduces the overall number of photons detected. This has to be clarified.

Related to my previous point 4, which is asking about the fact that the spectroscopic data is traded for 1D spatial data. The answer was:

"To clarify, the spectral measurement doesn't directly contribute to the resolution improvement. Instead, spectral measurement provides critical information for molecular discrimination. "

What does this mean?

"With this information, we can combine multiple blinking events emitted from the same molecule captured from different frames to achieve a higher localization precision as a result of increased photon counts, as shown in Fig. S11."

Again, I do not see the point about the photon count here.

Reply: SPLM improves the spatial resolution in the same manner as traditional PLM, i.e., to rely on larger photon counts. In each image frame, the photon counts are usually limited; however, if we can combine photon counts from the same molecule in different frames, the combined larger photon counts will contribute to the improved spatial resolution. The spectroscopic information in each blinking provides the criteria for such combination of photons from different image frames. More details are given below and in the Supplementary Note 2.

Conventional PLM fits a Gaussian kernel to the diffraction limited PSF and approximates the central location to improve imaging resolution. To give the most basic example, we first consider a single blinking molecule imaged by conventional PLM. After capturing a single blinking event

on the detector, depending on the detected photons, detector noise, pixel size, and other factors, the location of the single molecule can be approximated, with certain precision of the Gaussian kernel fitting. The localization precision depends on the photon counts in this particular frame. Since many molecules can blink multiple times during the imaging process, there is a chance that the same single molecule will blink at a later time and again be captured by the detector. In this scenario, the later blinking events from the sample molecule will be reconsidered by the PLM algorithm and will also be plotted to the image. However, since the parameters that were used to approximate the location of the single molecule have variation (different background, different photon yield, etc...), the probable location calculated by the Gaussian kernel fitting can be slightly different from the original location of the first blinking event.

If we know that those blinking events are coming from the same molecule, we will be able to combine photons from different frames to increase the total photon count, which further improves the localization precision. For instance, consider a single molecule which blinks a total of 16 times with a 200 detected photons in each blinking event. If we use only a signal frame for localization, the accuracy is determined by the photon count of 200. If we capture all the 16 blinking events from 16 different frames and combine them, the total combined photon count will be 3,200, which will improve the localization precision as a result of larger photon counts. However, in conventional PLM, it is hard to know whether these 16 blinking are from the same molecule. Therefore, the photon counts cannot be simply combined.

Detecting the spectral information of each blinking event provides the evidence for us to combine these 16 frames. SPLM offers exactly this capability. In SPLM, we use 25% of the photons for localization (zeroth order image) and 75% of the photons for spectra information (first order diffraction). In each frame, we have less photon counts ($200 \times 0.25 = 50$). But when we combine the 16 frames, the total photon counts will be $200 \times 0.25 \times 16 = 800$. In this case, we can accumulate a 4 times larger photon count. If there is a possibility to combine more frames, for example 160, the photon count can be enlarged further to 8000. Then the spatial resolution is improved using these enlarged photon count as compared with the 200 photon count in a single frame.

2, Also, it just occurred to me at the last reading that the authors define the spectral resolution purely through pixel sampling, giving 0.63 nm. Resolution and sampling are different quantities, because the spectral resolution is also affected by diffraction, it must depend on the properties of the grating (number of grooves illuminated), the distance to detector, etc. The resolution should be recalculated or measured.

Reply: We agree with this reviewer on the difference between "resolution and sampling". Our spectral resolution is indeed limited by the CCD spectral sampling. We calibrate the spectral resolution using a Mercury-Argon calibration lamp (SPL-HGAR, Photonics Technologies) with an entrance slit width of 10 μm , as shown in Supplementary Figure 3. By measuring the full width half maximum of the 546.08 nm peak, we calculated the spectral resolution to be 0.63 nm. This result indicates the system's spectral resolution is limited by the pixel size of the camera being used in our experiment, rather than the grating or the spectrometer.

We removed this value from the abstract and added more details to the Method section to

describe the spectral resolution of SPLM.

In Methods:

“We measured the system’s spectral resolution using a Mercury-Argon calibration lamp (SPL-HGAR, Photonics Technologies) with an entrance slit width of 10 μm , as shown in Supplementary Figure 3. By measuring the full width half maximum of the 546.08 nm peak, the spectral resolution was calculated to be 0.63 nm. This indicates the system’s spectral resolution is limited by the pixel size of the camera being used.”

Reviewer #3

[The referee believes the work is now acceptable for publication, with no further comments for the authors]

Reply: We greatly appreciate the reviewer’s efforts in carefully reviewing the initial submission and providing very helpful/valuable comments on how to improve our manuscript.